# Are Housing Prices Associated with Food Consumption?

**DOI:** 10.3390/ijerph17113882

**Published:** 2020-05-30

**Authors:** Edwin S. Wong, Vanessa M. Oddo, Jessica C. Jones-Smith

**Affiliations:** 1Department of Health Services, University of Washington, Seattle, WA 98195, USA; voddo@uic.edu (V.M.O.); jjoness@uw.edu (J.C.J.-S.); 2Nutritional Sciences Program, University of Washington, Seattle, WA 98195, USA

**Keywords:** housing prices, food consumption, obesity, dietary quality, economics

## Abstract

Objective: Since January 2010, the U.S. has experienced economic recovery, including a 39% increase in home prices nationally. While higher home prices represent a wealth increase for some homeowners, it may decrease real purchasing power for others. The objective of this study is to examine the relationship between local area housing values and consumption of four food categories. Design: Observational study using data from the Behavioral Risk Factor Surveillance System between 2011 and 2015. Outcomes included number of times per week food was consumed and binary measures denoting consumption ≥2 times per day for four categories: vegetables, fruit, legumes and fruit juice. The primary explanatory variables were metropolitan/micropolitan statistical area home and rental price indices from Zillow. Differential associations by home ownership, age, race/ethnicity and education were examined. Results: Overall, housing values were not associated with intake of vegetables or fruit juice. Among homeowners, a $10,000 increase in home price was associated with small, but statistically significant reductions in fruit and legume consumption. These inverse associations were pronounced among Hispanic and non-Hispanic Black adults. Conclusions: Lower fruit and legume consumption associated with greater housing values may represent one of several explanations including a decrease in purchasing power, given increases in home prices and limited wage growth since 2010.

## 1. Introduction

Food consumption and dietary quality have previously been associated with multiple economic factors. National data suggests that having lower income is associated with lower diet quality and expenditures on food [1]. An emerging literature has examined how economic conditions are associated with dietary quality, motivated by the onset of the Great Recession [2,3]. For example, several studies report that the recession was associated with lower consumption of fruits and vegetables, and higher consumption of energy-dense foods [4,5,6,7]. However, studies have yet to explore whether the post-Recession economic recovery was associated with food consumption, and whether potential associations vary by demographic characteristics and socioeconomic status (SES).

Although unemployment is the most commonly used measure to assess economic conditions, changes in housing prices and foreclosure rates also represent a significant shock to financial well-being. Previous studies have found that home foreclosure is associated with higher body mass index, greater rates of chronic disease and adverse health outcomes [8,9,10,11]. A small, but complementary literature suggests higher housing values are associated with improved weight-related outcomes [12,13]. Despite prior research, limited evidence elucidating the relationships between foreclosures and housing values, and food consumption currently exists. This is of interest given the substantial economic recovery in the United States (U.S.) since 2010, including a 39% increase in average home prices nationally.

An increase in housing prices has the effect of boosting housing wealth and overall lifetime income for households. Within the context of the permanent income hypothesis, individuals will adjust, or smooth expenditures over their lifetime according to their anticipated total lifetime income, and not current income [14]. In other words, greater housing wealth in the present increases an individual’s total lifetime financial resources, and may increase the quantity and quality of food consumed in the present. 

Higher housing prices may also represent a decrease in real purchasing power [15]. For younger households that need to borrow to purchase a home, higher housing prices may be reflected through higher monthly mortgage payments. Increasing housing prices may also diminish purchasing power for younger homeowners and renters seeking to purchase larger, more expensive houses in the future to accommodate a growing family or different life stage. For renters, decreasing purchasing power may result from higher monthly rental payments often linked to local area house prices. Households residing in homes with no housing debt, more common among older adults, do not face these decreases in purchasing power. Thus, we would anticipate the relationship between housing prices and food consumption to differ by age and home ownership status (i.e., renter vs. homeowner). 

The objective of this study was to examine the relationship between local area housing values and four categories of food consumption. Food consumption represents a measure of key public health interest. Understanding changes in this measure in relation to housing values, a key determinant of the macroeconomy, is important for surveillance efforts informing whether programs to promote healthy eating and food security are being reinforced or offset by the economic environment. In addition, economic recovery may yield unequal benefits between population subgroups defined by age, race/ethnicity and education. Evidence on differential relationships between these subgroups may provide more targeted insights for stakeholders in their surveillance efforts.

## 2. Materials and Methods 

### 2.1. Data Sources

The primary data source was the Behavioral Risk Factor Surveillance System, Selected Metropolitan/Micropolitan Area Risk Trends (BRFSS SMART) from 2011 through 2015. Administered by the Centers for Disease Control and Prevention, BRFSS is a health-related telephone survey that collects data on over 400,000 U.S. adults annually. BRFSS includes measures of health-related risk behaviors and events, chronic health conditions, and preventive service use; SMART data are a subset of selected metropolitan and micropolitan statistical areas (MMSAs). BRFSS SMART data were linked using MMSA identifiers with publicly available housing value data from Zillow.com and aggregate food price data from the Bureau of Labor Statistics. 

### 2.2. Study Sample

We identified 610,084 adults, age 18+, in BRFSS SMART datasets residing in 188 MMSAs during the period 2011 through 2015. MMSAs are geographical units generated by the U.S. Office of Management and Budget, which encompass a large population cluster and surrounding communities with substantial economic and social integration. MMSAs include both metropolitan (e.g., Seattle-Tacoma-Bellevue, WA, USA) and micropolitan (e.g., Pullman, WA, USA) statistical areas [16]. We excluded 445,339 adults who resided in MMSAs with missing food price index data, which was available for 25 MMSAs. After excluding 1034 adults with missing food consumption data for all categories examined, the final study sample consisted of 163,651 adults encompassing three distinct calendar years. 

### 2.3. Outcome Measures

Guided by the 2010 Dietary Guidelines for Americans, food consumption data in BRFSS was collected using responses to a 6-item brief dietary assessment, which ascertained intake of the following: dark green vegetables, orange vegetables, other vegetables, legumes, whole fruit and 100% fruit juice. For each category, respondents were asked how many times per day, week, or month food was eaten. We summed responses to the three vegetable categories into a single composite vegetable measure. In empirical analyses, we examined count measures denoting number of times food was eaten per week, and binary measures denoting intake of food at least twice per day. 

### 2.4. Explanatory Variables

The primary explanatory variable was MMSA-level housing value, measured using the Zillow Home Value Index, a seasonally adjusted measure of median home price calculated using several sources including prior sales, county records and tax assessments. MMSA-level home price indices were chosen to coincide with the smallest geographical unit available in BRFSS SMART data. MMSA-level home price indices were derived from the estimated market value for every home in an MMSA, more commonly known as “Zestimates.” The calculation of Zestimates accounts for the fact that different types of property may sell at greater rates at different periods of time (e.g., expensive homes selling at greater rates in the winter) [17,18]. We also examined MMSA-level home rental values, measured using the Zillow Home Rental Index, which is constructed using a similar methodology. Housing price indices are reported by Zillow on a monthly basis, which were averaged to an annual basis to coincide with person-year observations from BRFSS. 

### 2.5. Statistical Analysis

We used negative binomial regression with MMSA fixed effects to examine the relationship between housing values and frequency of food consumption. MMSA fixed effects account for time invariant area-level confounders correlated with both housing values and food consumption. Selection of negative binomial models addressed right skewness in the distribution of count measures of food consumption. We conducted unconditional estimation of fixed effects negative binomial models to address potential biases in other candidate models [19]. Similarly, we applied logistic regression with MMSA fixed effects to examine the relationship between housing values and consuming respective food categories at least twice per day. 

All regression models controlled for an MMSA-level aggregate food price index and individual demographics including age categories, gender, race/ethnicity (non-Hispanic (NH)-white, NH-black, NH-Asian, NH-other race, NH-multi-race, Hispanic/Latino), marital status, educational attainment (<high school, high school or General Education Development, some college, college graduate), income categories, home ownership status and year fixed effects. All analyses were weighted to the population of adults residing in MMSAs using BRFSS sampling weights that accounted for oversampling and non-response bias. Standard error estimates for model coefficients were heteroskedastic robust. 

For inference, we present average marginal effects (AMEs). For count measures of food consumption, AMEs represent the change in number of times per week a food category was consumed, attributable to a $10,000 increase in MMSA-level median home price. For binary measures, AMEs represent the change in probability of consuming food in a given category at least twice per day, attributable to a $10,000 change in home price. For analyses using MMSA-level home rental values, AMEs are with respect to a $100 change in median rental price. Standard errors for AMEs were calculated from coefficient estimates and recycled predictions using the delta method [20]. 

As described in the Introduction, economic theory posits the relationship between MMSA-level housing values (home or rental prices) and food consumption differs by home ownership status. Therefore, we interacted housing value variables with home ownership to examine these potential differential associations. We then generated interaction terms to assess differential associations by combinations of home ownership status and age. In an analogous manner, we explored potential differential associations across combinations of home ownership status and levels of SES defined by (1) race/ethnicity and (2) educational attainment. We present AMEs for MMSA-level housing values for homeowners and MMSA-level rental values for renters.

## 3. Results

### 3.1. Descriptive Statistics

Among BRFSS respondents over the three survey waves, the majority of individuals were female (51.5%), NH-white (58.9%) and owned a home (66.3%) (Table 1). Age was evenly distributed across the six age groups and similar proportions had some college experience (30.7%) or a college degree (29.5%). Homeowners were more likely to be at least 35 years of age (81.0% vs. 47.5%), NH-white (67.9% vs. 41.3%), have graduated college (35.2% vs. 18.4%), married (62.7% vs. 24.8%) and earned ≥$75,000 annually (38.1% vs. 10.7%). On average, home price and rental price in individuals’ MMSA of residence was $183,177 and $1452, respectively.

On average, individuals consumed vegetables, fruits, legumes and fruit juice 10.7 (SD = 9.4), 7.0 (SD = 8.0), 2.0 (SD = 4.1) and 1.9 (SD = 3.0) times per week, respectively. Less than one-third of individuals consumed each respective category at least twice per day. Compared to renters, homeowners consumed vegetables and fruit more often, but consumed legumes and fruit juice less often in a given week. 

### 3.2. Relationship between Housing Value and Food Consumption, by Home Ownership

Among all homeowners, a $10,000 increase in MMSA-level home price was not significantly associated with the frequency of food consumed per week for each of the four categories examined (Table 2). However, a $10,000 increase in home price was associated with a 0.03 percentage point (pp) decrease (*p* = 0.039) in the probability of eating fruit and a 0.02 pp decrease (*p* = 0.006) in the probability of eating legumes at least twice per day. MMSA-level rental prices were not associated with all four categories of food examined in this study. 

### 3.3. Housing Values and Food Consumption Among Homeowners, by Sociodemographics

#### 3.3.1. Age

MMSA-level home prices were inversely associated with the probability of consuming fruit ≥2 times per day among age 25–34 (AME = −0.46 pp, *p* = 0.015), and age 65+ (AME = −0.34 pp, *p* = 0.030), individuals, respectively (Figure 1). Home prices were also inversely associated with frequency of legumes per week only for age 65+ individuals (AME = −0.027, *p* = 0.032), and similar statistically significant declines in the probability of consuming legumes ≥2 times per day across nearly all age groups of approximately 0.2 pps. Higher home prices were associated with fewer times fruit juice was consumed only among the youngest age group (AME = −0.039, *p* = 0.024). We did not identify statistically significant associations between home prices and vegetable consumption across all age groups (Appendix A).

#### 3.3.2. Race/Ethnicity

The magnitude of the association between higher home prices and lower consumption of fruit was greater among non-whites (Figure 2). Specifically, a $10,000 increase in MMSA-level home price was associated with 0.085 (*p* = 0.012), 0.080 (*p* = 0.019) and 0.109 (*p* = 0.026) fewer times per week fruit was consumed for NH-black, Hispanic, and NH-other race adults, respectively. In contrast, home prices were not associated with frequency of fruit consumption for NH-white adults. We found analogous declines in the probability of consuming fruit ≥2 times per day for NH-black (AME = −0.65 pp, *p* = 0.001), Hispanic (AME = −0.40 pp, *p* = 0.036), and NH-other race adults (AME = −0.69 pp, *p* = 0.016), which was greater in magnitude compared to declines for NH-white adults (AME = −0.35 pp, *p* = 0.021). For legumes, higher home prices were associated with a lower frequency of consumption per week only for Hispanic adults (AME = −0.041, *p* = 0.013). However, higher home prices were associated with a decrease in the probability of consuming legumes ≥2 times per day for NH-white (AME = −0.08 pp, *p* = 0.023), NH-black (AME = −0.14 pp, *p* = 0.027), Hispanic (AME = −0.25 pp, *p* = 0.027) and multi-race (AME = −0.34 pp, *p* = 0.009) adults. 

#### 3.3.3. Education

The relationship between higher home prices and lower fruit consumption was statistically significant only among adults who graduated from college for both frequency per week (AME = −0.069, *p* = 0.034) and probability of consuming ≥2 times per day (AME = −0.43 pp, *p* = 0.015) (Appendix A). Higher home prices were associated with a lower probability of consuming legumes ≥2 times per day, with a similar magnitude across levels of education. 

### 3.4. Housing Values and Food Consumption Among Renters, by Sociodemographics

#### 3.4.1. Age

Among renters, the absence of a statistically significant association between rental prices and food consumption was largely present in the analysis of differential effects (Appendix A). However, a $100 increase in rental price was associated with a reduction in frequency of eating fruit of 0.232 times per week (*p* = 0.005) among age 65+ adults.

#### 3.4.2. Race/Ethnicity

Higher rental prices were associated with a lower frequency of fruit consumed per week for Hispanic (AME = −0.303, *p* = 0.001) and NH-other race adults (AME = −0.267, *p* = 0.031), as well as the probability of consuming fruit ≥2 times per day for Hispanic adults (AME = −1.020 pp, *p* = 0.035). For fruit juice, higher rental prices were associated with a reduction in frequency of consumption per week (AME = −0.525 times per week, *p* = 0.043) among NH-white adults. 

#### 3.4.3. Education

The analysis of differential effects by education indicated higher rental prices were associated with a reduction in the number of times fruit (AME = −0.210, *p* = 0.010) and fruit juice (AME = −0.069, *p* = 0.016) were consumed per week among adults completing college. 

## 4. Discussion

Theory posits that economic factors, such as income and wealth resources, are important determinants of food consumption. Recent research has examined the effects of the Great Recession on dietary outcomes, however, there is limited evidence informing the effects of the post-recession economic recovery. In addition, few studies have examined how increases in housing values, characteristic of the post-recession recovery in the U.S., influence consumption of different categories of food. This is significant, as the rising cost of housing associated with economic recovery may crowd out individuals’ spending on healthy foods and offset public health efforts to promote healthy eating. 

Findings indicate variation in consumption of the four categories of food examined in this study. Notably, the percentage of individuals consuming a given food category ≥2 times per day ranged from 2.0% for legumes to 26.7% for vegetables. Among homeowners overall, home prices were not associated with weekly frequency of food consumed for each category. However, estimates across all homeowners masked a statistically significant inverse relationship between housing prices and consumption of fruit among NH-non-white adults. Marginal effects represented a decrease of between 1.1% (i.e., −0.080/7.3) to 1.5% (i.e., −0.109/7.3) of average weekly fruit consumption by homeowners. We also identified small, but statistically significant associations between home prices and weekly consumption of legumes among respective groups of Hispanic and age 65+ individuals, and fruit juice among age 18–24 individuals.

Among all homeowners, higher MMSA-level home prices were associated with modestly lower probabilities of consuming fruits and legumes at least two times per day. For fruit, pronounced associations among NH-black and NH-other race adults, and college graduates coincided with results for weekly frequency of consumption. We also identified pronounced associations for fruit consumption among age 25–34 and 65+ subgroups. For legumes, statistically significant associations were pronounced for age 18–24 and Hispanic and multi-race subgroups.

For renters, housing value was not associated with each of the four food categories overall. However, in the analysis of differential effects, we found the inverse association between rental prices and food consumption was present for fruit among Hispanic and multi-race individuals, as well as those age 65+ and college graduates.

Identified changes in fruit consumption were surprising given prior research has found fruit is a normal good (i.e., consumption increases as income increases) [21,22]. One potential explanation for the inverse relationship between fruit consumption and housing values is the reduction in purchasing power accompanying increasing housing values. Reductions in purchasing power may be present given that housing values have increased, while wage income has exhibited little growth since 2010 [23]. Findings in subgroup analyses provide some support for the purchasing power hypothesis. Declines in fruit consumption attributable to housing price identified in this study were pronounced among racial/ethnic minorities who have fewer financial resources at a population level. Marked declines were also present among the oldest adults, who may have less disposable income, relative to middle age adults in their prime working years. Between 2010 and 2017, 36% of individuals earning less than $30,000 annually owned a home [24].

Fruit results may also be driven by the definition of the BRFSS survey measure, which asks about the number of times fruit was consumed, independent of serving size or quality (e.g., organic, free trade, etc.). As a result, reductions in fruit consumption could reflect larger serving sizes consumed less often, or higher quality fruit consumed less often. The fruit survey measure also asks respondents to consider fresh, canned and frozen sources collectively. Thus, reductions in fruit consumption identified could reflect changes within subcategories resulting in an overall net decline in fruit consumption overall (e.g., increase in fresh fruit together with a larger decrease in canned fruit).

Another key finding was a robust inverse association between housing value and intake of legumes. This finding is consistent with at least two potential explanations. First, prior research has identified legumes as an inferior good (i.e., consumption decreases as income increases) [25]. If greater housing values represent a positive wealth effect, then individuals may be consuming alternatives to legumes as a source of protein such as meat. Second, similar to fruit consumption, the inverse association between housing values and legumes may reflect reductions in real purchasing power. Contextual information on legume consumption combined with results from subgroup analyses provide some support for this explanation. Specifically, legumes are often considered a staple item of low-income households and in this study, among adults reporting consumption of legumes at least two times per day, 44% earned <$25,000 annually (compared to 24% of all adults). This was accompanied by inverse associations between housing values and legume consumption being pronounced among respective subgroups of Hispanic, multi-race and low education homeowners, who have lower financial resources at the population-level relative to other subgroups. Within the purchasing power hypothesis, a less consistent inverse relationship between housing values and fruit and legume consumption among renter subgroups may be due to direct homeownership costs such as property tax and home insurance not faced by renters. It should be noted that reductions in both fruit and legumes attributable to changing housing values are small in magnitude, and the influence of purchasing power or other mechanisms is likely to be modest. 

This study has several limitations. First, food consumption measures were derived from self-report and captured the number of times an individual consumed a general food category over a specified time interval. This measure of food intake does not account for serving size and may be subject to recall bias. Second, we controlled for a composite food price index to account for changes in food prices correlated with housing values. However, estimated associations between housing prices and food intake may be biased if the composite index does not reflect variation in prices for individual food categories. Third, we examined consumption patterns of Americans residing in a sample of all MMSAs in the U.S. Thus, findings may not generalize individuals across all areas, including those residing in rural areas.

## 5. Conclusions

Over a five-year period following the Great Recession, increases in housing values were associated with small reductions in consumption of fruits and legumes. Identified associations were pronounced in several subgroups including racial/ethnic minorities. Several potential explanations exist for our findings including reductions in purchasing power for some individuals accompanying rising housing values. Taken together, findings suggest greater housing values may slightly offset public health efforts to increase consumption of healthy food such as fruit and legumes for some individuals. Future research should examine how changes in more granular measures of food consumption including specific food types and nutrient intake to provide a more complete picture of the effects of the macroeconomy.

## Figures and Tables

**Figure 1 ijerph-17-03882-f001:**
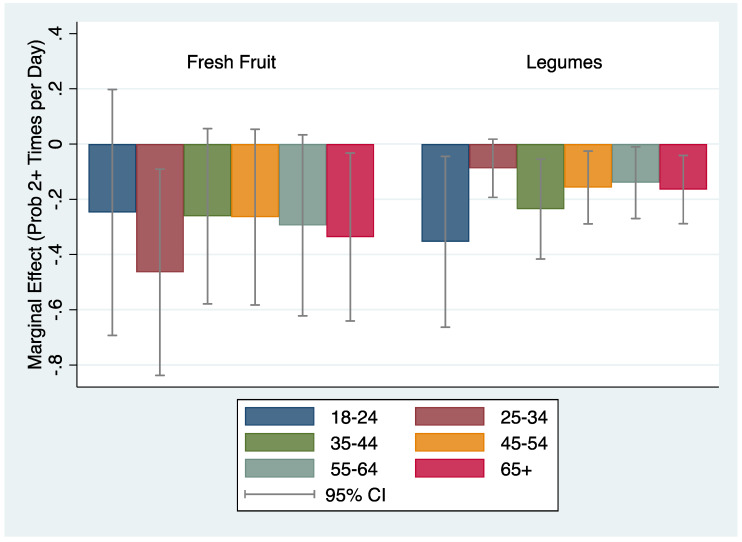
Differences in the association of housing price and food consumption by age group.

**Figure 2 ijerph-17-03882-f002:**
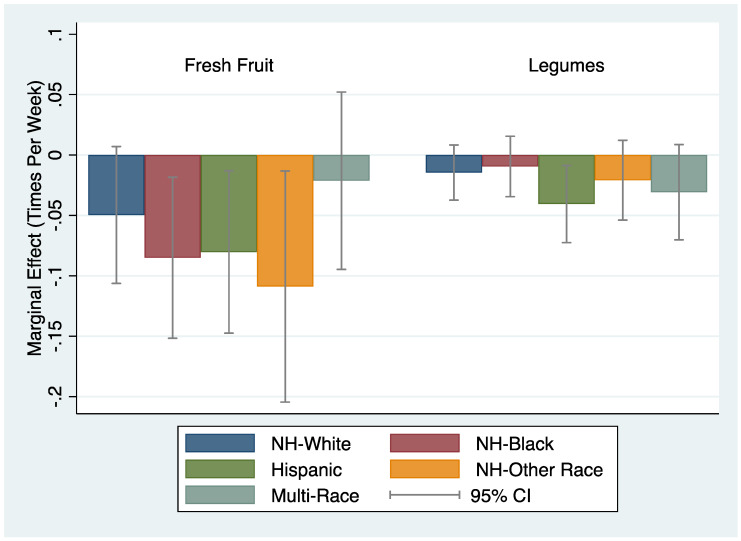
Differences in the association of housing price and food consumption by race/ethnicity.

**Table 1 ijerph-17-03882-t001:** Selected descriptive statistics of study sample, stratified by home ownership status.

	All Adults	Homeowners	Renters
*N*	163,651	117,798	45,853
Weighted *N*	132,923,180	88,166,094	44,757,087
MMSA Home Price ^1^ (mean/sd)	183.2 (69.9)	181.1 (70.5)	187.2 (67.4)
MMSA-level Rental Price ^2^ (mean/sd)	1452 (279)	1440 (287)	1474 (259)
Food Price Index (mean/sd)	236.1 (15.5)	236.3 (16.2)	235.8 (14.1)
Age Group (%)			
18–24	12.2	6.7	23.1
25–34	18.1	12.4	29.4
35–44	17.8	18.0	17.6
45–54	18.8	21.6	13.3
55–64	16.0	19.6	8.9
65+	17.1	21.8	7.7
Female (%)	51.5	52.3	49.7
Race/Ethnicity (%)			
NH White	58.9	67.9	41.3
NH Black	14.5	10.9	21.8
Hispanic	5.7	5.2	6.7
Other Race	1.2	1.0	1.6
Multi-Race	17.8	13.5	26.3
Missing	1.9	1.6	2.4
Education (%)			
< High School	13.6	9.4	21.9
HS Graduate	25.8	24.3	28.7
Some College	30.7	30.9	30.4
College Graduate	29.5	35.2	18.4
Missing	0.3	0.2	0.6
Marital Status (%)			
Married	49.9	62.7	24.8
Divorced	10.5	9.5	12.5
Widowed	6.6	7.7	4.6
Separated	2.6	1.5	4.7
Never Married	24.8	14.9	44.2
Unmarried Couple	4.9	3.3	8.1
Missing	0.6	0.4	1.0
Income Category (%)			
<$10,000	5.4%	2.2%	11.6
$10,000–$14,999	4.4%	2.4%	8.5
$15,000–$19,999	6.5%	4.1%	11.2
$20,000–$24,999	7.8%	5.9%	11.5
$25,000–$34,999	9.1%	7.9%	11.5
$35,000–$49,999	11.5%	12.0%	10.6
$50,000–$74,999	13.1%	15.4%	8.6
≥$75,000	28.9%	38.1%	10.7
Missing	13.3%	12.0%	15.9
Own Home (1 = yes) (%)	66.3	100.0	0.0
# Times per Week (mean/sd)			
Vegetables	10.8 (9.4)	11.3 (9.2)	9.7 (9.6)
Fruit	7.0 (8.0)	7.3 (7.8)	6.3 (8.1)
Legumes	2.0 (4.1)	1.9 (2.9)	2.1 (5.3)
Fruit Juice	2.0 (3.0)	1.9 (2.9)	2.1 (3.0)
>2 Times per Day (%)			
Vegetables	26.7	28.6	23.0
Fruit	21.8	23.4	18.5
Legumes	2.0	1.6	2.7
Fruit Juice	5.0	4.1	6.9

MMSA = metropolitan/micropolitan statistical area, NH = non-Hispanic, HS = high school. ^1^ Dollars in thousands. ^2^ Monthly rent price in dollars.

**Table 2 ijerph-17-03882-t002:** Marginal effects of housing value on consumption of four food categories, stratified by home ownership status.

	Renters				Homeowners			
	All Veg	Fruit	Legume	Juice	All Veg	Fruit	Legume	Juice
# times per week								
Housing Value ^1^	−0.028	−0.108	−0.011	−0.055	0.001	−0.005	−0.002	−0.001
	(0.088)	(0.068)	(0.029)	(0.029)	(0.004)	(0.003)	(0.001)	(0.001)
>2 times per day								
Housing Value ^2^	−0.12	0.01	0.03	0.01	−0.01	−0.03 *	−0.02 **	−0.01
	(0.39)	(0.34)	(0.16)	(0.29)	(0.02)	(0.02)	(0.01)	(0.01)

* *p* < 0.05, ** *p* < 0.01. Veg = vegetables. ^1^ Change in times per week a food category is consumed attributable to a $100 increase metropolitan/micropolitan level home rental prices (for renters) or a $10,000 increase in home price (for homeowners). ^2^ Change in the probability of consuming a food category ≥2 times per day attributable to a $100 increase metropolitan/micropolitan (MMSA) level home rental prices (for renters) or a $10,000 increase in home price (for homeowners).

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
