# Peer review of "Are Housing Prices Associated with Food Consumption?"

_ijerph, 2020, doi:10.3390/ijerph17113882_

Round 1

Reviewer 1 Report

In this paper, the Authors have studied the relationships between housing values and food consumption, with specific reference to four food typologies. By using negative binomial/logistic regressions, they found that a higher housing value corresponds to a lower food consumption, with differences in the marginal effects related to the four food typologies considered.

The research must be improved and deepened.

The econometric methods applied are only recalled, whereas the models obtained for the different implementations are not reported. Furthermore, an accurate analysis of the statistical performances of the models is missing.

Moreover, the innovative contribution of the research is not clear, and the reliability of the outputs are not demonstrated. The Conclusions are very poor, outlining that the work constitutes an interesting application, but in the current version it does not adequately fit the aims of the Journal and consequently the interests of the potential readers.

Reviewer 2 Report

%MCEPASTEBIN%%MCEPASTEBIN%

The authors present an analysis of the BRFSS dataset between 2011-2015, investigating the association between diet quality and housing prices. They find no association of housing values with the frequency of vegetables or fruit juice consumption, but a significant small association with fruit and legume consumption (especially pronounced among Hispanic and non-Hispanic black adults). They maintain that these inverse associations may be explained by a decrease in purchasing power over time.

Comments:

  1. It is possible that a negative association between housing prices and food consumption could have some alternative explanations, such as:
    1. Income elasticity of demand consideration: Is it possible that lower consumption of legumes is actually a signal of higher diet quality? For example, if canned beans are an inferior good (demand decreases as income increases), then people with higher income (who are also home owners) will substitute away from canned beans towards more expensive food alternatives.
    2. Food variety has increased over time: thus, people with higher income may be purchasing more expensive alternatives (e.g. organic fruit or quinoa) in smaller quantities, rather than, for instance, bananas / white rice / canned beans (which could be purchased in higher quantity). If this is not possible to control for in the analysis, authors may want to consider whether this could be an additional explanation to their results.
  2. Authors find no overall association of food intake and rental prices, but a negative association of fruit/legume intake and house prices. I wonder if they could elaborate on what they believe is driving this difference. If purchasing power is, indeed, the explanation for the negative association, shouldn’t we be seeing more of this in people who rent expensive housing?
  3. Authors seem to use the term “servings” and “times per day” interchangeably. However, the frequency of food consumption (“times a day”) and amount of consumption (“servings a day”) are not the same concepts. For example, the amount may be higher when one consumes a large salad (1 time per day) versus three apple slices (3 times a day). While authors refer to “servings” consumed per day, it is unclear from their methods, whether they indeed converted the BRFSS reported “times per period” to “servings per period” (see the following link for an example of an algorithm created for this conversion: https://academic.oup.com/aje/article/181/12/979/91630 ). Please specify, whether this conversion was done, and if not – please rephrase “servings” or “amount” consumed to “times” or “frequency” consumed.
    1. The paper in the link above also contains SAS code for conversion purposes (see supplemental files)

Round 2

Reviewer 1 Report

The main indications have been taken into account.